# Modulating Cell–Scaffold Interaction via dECM-Decorated Melt Electrowriting PCL Scaffolds

**DOI:** 10.3390/polym17233133

**Published:** 2025-11-25

**Authors:** Wenchao Li, Xiang Gao, Peng Zhang

**Affiliations:** 1School of Advanced Manufacturing, Nanchang University, Nanchang 330031, China; 2School of Stomatology, Jiangxi Medical College, Nanchang University, Nanchang 330031, China; 3Jiangxi Provincial Key Laboratory of Oral Diseases, Nanchang 330031, China; 4Jiangxi Provincial Clinical Research Center for Oral Diseases, Nanchang 330031, China

**Keywords:** melt electrowriting (MEW), decellularized extracellular matrix(dECM), polycaprolactone (PCL), decoration

## Abstract

Aligned fibrous scaffolds are essential for directing soft-tissue regeneration, yet synthetic polymers lack native biochemical cues. To bridge this gap, bioactive and anisotropic scaffolds were developed by combining melt electrowriting (MEW) with decellularized extracellular matrix (dECM) decoration to enhance cell–scaffold interactions for soft tissue engineering. Porous polycaprolactone (PCL) scaffolds with aligned microfibers and tunable pore architectures (aspect ratios 1:1, 1:2, and 1:3) were fabricated via MEW and subsequently coated with porcine skeletal muscle dECM using a dip-gelation method. Comprehensive surface characterization confirmed the presence and robust adhesion of the dECM coating on the PCL scaffolds, which concurrently enhanced surface hydrophilicity. Furthermore, mechanical testing demonstrated that the resulting composite scaffold retained the structural integrity required to meet the mechanical demands of tissue regeneration. In vitro studies using L929 fibroblasts demonstrated that dECM decoration significantly improved cell adhesion, proliferation, and alignment along the fiber direction. Notably, scaffolds with 1:1 and 1:2 aspect ratios supported the highest cell density and guided morphological elongation most effectively. These findings highlight the synergistic potential of topographical cues and biochemical signaling in scaffold design for functional tissue regeneration.

## 1. Introduction

In the oral and maxillofacial region, soft tissue injuries to muscles, nerves, and blood vessels can severely impair crucial functions like mastication, speech, and facial expression [1]. The proper functioning of these tissues is highly dependent on their orderly cellular architecture [2]. For instance, in vascular tissue, endothelial cells align with blood flow, which is critical for regulating antithrombotic responses and biological signaling [3]. Therefore, highly oriented cell arrangement is of great significance for developing functional in vitro models aimed at repairing oral and maxillofacial soft tissues.

Typical tissue engineering strategies focus on designing scaffolds that mimic the native extracellular matrix (ECM) to guide cells into physiologically relevant 3D structures [4]. Anisotropic scaffolds, in particular, can simulate the aligned morphology of natural ECM, effectively directing cell arrangement, elongation, proliferation, and differentiation [5]. These structures usually influence cellular behavior through mechanical regulation of the cytoskeleton or by establishing physicochemical gradients, thereby promoting organized tissue formation. Studies have shown that anisotropic porous, microfibrous, or nanofibrous scaffolds can enhance the alignment, differentiation, and myofibrillar organization of cells [6]. Compared with scaffolds with random fibers, scaffolds with arranged fibers can better promote the arrangement of Schwann cells [7]. Techniques such as melt electrowriting (MEW) [8], electrospinning (ESP) [9], and fused deposition modeling (FDM) [10] have been widely used to fabricate aligned fibrous scaffolds with high porosity and tunable fiber diameters. For example, MEW allows precise fiber placement at the micrometer scale and has been used to create highly aligned structures that guide cell behavior [11]. Despite offering exceptional micro-architectural guidance, the inherent bio-inertness of synthetic MEW scaffolds often presents a less favorable environment for cell adhesion. To address this limitation, surface functionalization with bioactive coatings, such as gelatin and yeast-derived peptides, has emerged as an effective strategy to enhance the overall biocompatibility of MEW scaffolds by fostering more favorable cell-scaffold interactions [12,13].

Decellularized extracellular matrix (dECM) is a highly bioactive material derived from native tissues, capable of providing tissue-specific biochemical and physical cues to establish a supportive microenvironment that significantly promotes cell attachment, proliferation, and functional differentiation [14]. Harnessing this potential, Junka et al. combined a polycaprolactone (PCL) scaffold with chondrocyte-derived dECM, calcium phosphate, and albumin to create a composite fibrocartilage-mimetic scaffold designed to emulate a woven bone precursor [15]. Sharma et al. fabricated 3D nanofiber scaffolds via gas-foaming expansion and functionalized them with cell-derived dECM to create a highly bioactive interface that enhances cellular alignment, reduces immune response, and improves tissue regeneration [16]. Despite this progress, the focus has predominantly been on scaffolds composed of nano- or sub-microscale fibers, such as those produced by electrospinning. The surface modification of melt electrowriting (MEW) scaffolds—which feature precisely controlled fiber diameters in the tens of micrometers—has received comparatively limited and unsystematic attention. An alternative integration strategy involves encapsulating the MEW scaffold within hydrogel [17]. In this configuration, the MEW structure acts primarily as a mechanical support skeleton, while the hydrogel provides a bioactive environment for cell growth. In contrast to surface functionalization, this method often fails to leverage the finely tuned micro-architectural guidance of the MEW scaffold, as the critical topographical cues of the fibers are obscured by the surrounding hydrogel matrix.

Therefore, this work aims to develop a biomimetic scaffold that synergistically integrates the topographic guidance of structurally anisotropic MEW PCL scaffolds with the biochemical signaling of dECM through a surface decoration approach. The ultimate goal is to systematically investigate cellular responses, including adhesion, proliferation, and alignment, to this dual-functional platform. To achieve this, dECM-decorated PCL scaffolds with controlled pore architectures (aspect ratios of 1:1, 1:2, and 1:3) were fabricated via MEW combined with a dip-gelation method. The biological performance of these composite scaffolds was thoroughly evaluated, focusing on their effects on fibroblast adhesion, viability, proliferation, and morphology. This study provides fundamental insights into the interplay between topographical and biochemical cues in guiding cell behavior, thereby laying a necessary foundation for the future development of advanced scaffolds for muscle tissue regeneration.

## 2. Materials and Methods

### 2.1. Decellularization of Porcine Skeletal Muscle Tissue

Fresh skeletal muscle samples were obtained from the lower limbs of healthy pigs purchased from a commercial supermarket. The fascia and obvious connective tissues were removed, and the isolated skeletal muscles were immersed in PBS (Aladdin Reagent (Shanghai) Co., Ltd., Shanghai, China) containing 1% (wt/v) antibiotic/antifungal agent, stirred at low temperature for 24 h (4 °C, 100 r/min), and washed in PBS. For the decellularization process, the skeletal muscle (outer muscle layer) was removed, and the muscle tissue was cut into 2 × 2 × 2 mm^3^ size, and then the treated muscle tissue was decellularized. Briefly, the muscle tissues were rinsed 3 times (5 min each) in Dulbecco’s phosphate-buffered saline (DPBS) c and stirred in PBS buffer containing 0.025% (wt/vol) trypsin and 0.05% (wt/vol) EDTA for 1h at room temperature (100 r/min), and then washed 3 times in PBS before being stirred in 1% Triton X-100 solution containing 1% antibiotics at the tissues were stirred at 4 °C for 24 h. The tissues were stirred in alternating hypotonic [10 mM tris-HCl] and hypertonic [50 mM tris-HCl and 1.5 M NaCl] salt solutions for 30 min each at room temperature, and the solutions were changed alternately three times until the muscle tissues became clear or white, and the muscle tissues were stirred at low temperature for 24h in ultrapure water and washed three times with PBS to remove the residual chemicals. The obtained dECM was frozen overnight in a −80 °C refrigerator and freeze-dried twice in a lyophilizer (Labconco, Kansas, MO, USA) for 12 h. The freeze-dried decellularized tissues were ground into a powder, and the dECM was digested by using 0.1% (wt/vol) pepsin in 0.01 Mhcl solution (1 mL of pepsin solution per 10 mg of muscle tissue). The prepared solution was stirred continuously for 48 h at room temperature, and the dECM solution obtained after digestion was stored in a −20 °C refrigerator until use. All chemical reagents were obtained from Xavier Biologics (Wuhan, China) unless stated otherwise.

### 2.2. Characterization of dECM

To observe the microstructure of decellularized tissues, natural and decellularized muscle tissues were soaked in 10% neutral buffered formalin (NBF, Leica Biosystems, Buffalo Grove, IL, USA) for 48 h at room temperature [18]. After fixation, the tissues were washed three times with PBS and gradually dehydrated using different ratios of ethanol solutions (30%, 50%, 70%, 80%, 90%, 95%, and 100%) and freeze-dried. The dried samples were sputter plated with gold (E-1045, Hitachi High-Technologies Naka Office, Ltd., Japan) and observed using a scanning electron microscope (JSM-IT700HR, JEOL Ltd., Japan). For histological analysis, natural and decellularized muscle tissues were fixed with 10% NBF for 48 h at room temperature. Section samples of 5 μm thickness were then stained with hematoxylin and eosin (H&E), Masson’s Trichrome, and 4′,6-diamidino-2-phenylindole (DAPI, (KeyGEN BioTECH, Nanjing), respectively, and the stained samples were observed using a Pannoramic250 digital section scanner manufactured by 3DHISTECH (Hungary). To measure the DNA content of the dried decellularized tissue, the water content of the decellularized tissue was measured. The samples were then analyzed biochemically to determine the DNA content in the dried samples. For DNA quantification, DNA was extracted from natural and decellularized skeletal muscle tissues using the Animal Tissue DNA Isolation Kit (manufactured by FOREGENE, Inc.) (Enzymatic Instrument, model, SpectraMAX Plus384, manufactured by Meigu Molecular Instruments, Inc.) to measure the DNA content of the samples. Then, quantitative biochemical analysis of major ECM components (collagen, GAGs, and elastin) was performed on native and decellularized tissues. Following a standardized protocol, 20 mg of freeze-dried tissues were solubilized using specific chemical reagents: pepsin in 0.5 M acetic acid for collagen, a papain solution for GAGs, and 0.25 M oxalic acid for elastin. The resulting solutions were then analyzed using dedicated commercial assay kits (CUSABIO^TM^ for collagen I, Sangon Biotech^TM^ for GAGs, and BioSS^TM^ for elastin; Wuhan Servicebio Technology Co., Ltd, China) in strict accordance with the manufacturer’s instructions.

### 2.3. Preparation of dECM/PCL Scaffold

PCL (Solvay, USA; average molecular weight: 80,000 Da) was used as the feedstock for melt electrowriting (MEW) [19]. The process was carried out on a customized MEW system, comprising a three-axis (X, Y, Z) precision stage, a custom printing nozzle, an optical monitoring system, a high-voltage power supply (DW-P303-1ACF0, Dongwen High Voltage, China), and an air pressure pump. During printing, the molten PCL was extruded through the nozzle under applied air pressure. A high-voltage power supply was used to electrically charge the melt, which—under the combined effects of electrostatic stress, extrusion pressure, viscous force, surface tension, and gravity—formed a Taylor cone at the nozzle tip. A potential difference of 4.0–4.1 kV between the nozzle and the collector plate generated a stable polymer jet, which was deposited onto the plate positioned at a working distance of 2.3 mm. The printing process was monitored in real-time using a high-resolution camera. Scaffolds with grid aspect ratios of 1:1, 1:2, and 1:3 (corresponding to unit cell dimensions of 400 μm × 400 μm, 400 μm × 800 μm, and 400 μm × 1200 μm, respectively) were fabricated according to the predefined patterns. The overall dimensions of all PCL scaffolds were 12 mm × 12 mm × 1 mm.

The neutralized dECM solution (1 mg/mL) was placed in an ice-water mixture to prevent physical gelation. The PCL scaffolds were first subjected to 5 M NaOH ethanol solution (Aladdin Reagent (Shanghai) Co., Ltd., Shanghai, China) for 10 min to enhance their hydrophilicity [20]. Subsequently, neutralization and cleaning thoroughly pretreated scaffolds were fully immersed in the dECM solution and placed under a brief vacuum to ensure thorough infiltration of the matrix into the porous structure. Following immersion, the scaffolds were withdrawn from the solution at a controlled, slow rate to achieve a uniform coating. Finally, to stabilize the deposited dECM layer, the scaffolds underwent a physical crosslinking process by being frozen and lyophilized (Labconco, Kansas, MO, USA), thereby securing the biofunctional coating onto the fiber surfaces [21] (Figure 1).

### 2.4. Mechanical Testing

Scaffolds fabricated with different porosity structures were subjected to uniaxial tensile testing on a universal testing machine (Meters Industrial Systems Ltd., Shanghai, China). The test was performed on 20-layer specimens at room temperature, with a clamp distance of 10 mm and a crosshead speed of 15 mm/min. The tensile strength and Young’s modulus were calculated based on the acquired stress–strain curves.

### 2.5. In Vitro Bioactivity of dECM Based Scaffolds

To evaluate the effect of PCL scaffold composition on fibroblast behavior, L929 cells were seeded onto the scaffolds and cultured for 3 days. Following culture, the samples were fixed in 4 wt% paraformaldehyde, permeabilized, and stained for nuclei (DAPI, KeyGEN BioTECH, Naning, China) and F-actin (TRITC-phalloidin, MKBio, Shanghai, China). Cell proliferation and alignment were then assessed using fluorescence microscopy (Leica DMi8, Leica Microsystems, Germany).

### 2.6. CCK8

The cytotoxicity of the fabricated scaffolds was evaluated using a Cell Counting Kit-8 (CCK-8, Beyotime, Shanghai, China) assay. Briefly, L929 fibroblasts (obtained from the Chinese National Collection of Authenticated Cell Cultures) were seeded onto the samples placed in a 12-well plate and incubated at 37 °C in a humidified atmosphere containing 5% CO_2_. After culturing for the designated periods, 1 mL of fresh medium containing 100 μL of CCK-8 reagent was added to each well. Following 1 h of incubation, the absorbance of the solution in each well was measured at 450nm using a spectrophotometer (Infinite 200 Pro, Tecan). Cell viability was calculated according to the following formula:(1)$$\mathrm{Cell}\;\mathrm{viability}=\frac{\mathrm{OD}\;\mathrm{treated}\;-\;\mathrm{OD}\;\mathrm{free}}{\mathrm{OD}\;\mathrm{control}\;-\;\mathrm{OD}\;\mathrm{free}}\;\times \;100\%,$$
where OD_treated is the absorbance of the wells containing cells and scaffolds, OD_control is the absorbance of the wells containing cells only (control group), and OD_blank is the absorbance of the wells containing only culture medium (blank group).

### 2.7. Surface Morphology and Characterization with SEM, FTIR and Contact Angle

The sample preparation protocol for scanning electron microscopy (SEM) differed depending on the presence of cells. For dECM/PCL scaffolds with cells, a standard protocol was followed to preserve cellular structures. The samples were washed with 0.1 mol/L PBS, fixed sequentially with 3% glutaraldehyde and 1% osmium tetroxide (each for 1 h), and dehydrated in a graded ethanol series [22]. In contrast, pure PCL and dECM/PCL scaffolds without cells did not require this fixation and dehydration process. All samples were sputter-coated with gold prior to imaging. Morphological observation was performed using an SEM (JSM-IT700HR, JEOL, Japan), with images captured from overall, top, and cross-sectional perspectives to characterize the surface and internal architecture.

The functional groups of the PCL and dECM/PCL composite scaffold were confirmed using Fourier transform infrared spectroscopy (FTIR; Thermo Scientific, USA) in a range of 4000–500 cm^−1^. The hydrophilicities of the PCL and composite scaffolds were determined with a contact angle goniometer (JC2000D3, Powereach, equipped with CCD camera and imaging software) at room temperature. For each measurement, a droplet of deionized water (4 μL) was pipetted onto the scaffold surface. Images of the water droplet were taken with a high-speed digital camera, and the contact angle values were calculated from these images.

### 2.8. Statistical Analysis

All data were expressed as mean ± standard deviation for at least three independent experiments. The differences between the control and experimental groups were compared using the Kruskal–Wallis one-way analysis of variance (ANOVA) and *t*-test to calculate statistical significance (GraphPad prism 8.0.1).

## 3. Results and Discussion

### 3.1. Decellularization of Skeletal Muscle Tissue

Decellularization aims to remove cellular components from native tissues through mechanical or chemical means, yielding an extracellular matrix (ECM)-rich material. This study employed an efficient protocol to decellularize skeletal muscle [18]. The decellularized ECM (dECM) displayed a translucent, whitish appearance with visibly enlarged inter-fiber gaps, attributable to the removal of components such as blood and connective tissues (Appendix A). This structural change was corroborated by SEM (Figure 2A,E), which showed a reduced myofiber density in the dECM. Subsequently, histological analyses with H&E and DAPI staining affirmed the near-complete absence of nuclei (blue, Figure 2D,H), while the preservation of the cytoplasmic component (pink, Figure 2B,F) was evident in H&E images [23]. This was further corroborated by Masson’s trichrome staining, which revealed well-preserved muscle fibers (red) alongside minimal residual collagen components (blue, Figure 2C,G) [24]. Collectively, these results confirmed the effective removal of cellular material and the successful retention of key extracellular matrix structures.

To quantitatively evaluate decellularization efficiency, DNA content was analyzed relative to tissue dry weight. A significant decrease in DNA content was observed in the dECM (13.06 ± 1.47 ng/mg) compared to natural muscle (71.83 ± 3.10 ng/mg) (Figure 3A), confirming the effective removal of cellular material, such as nuclear components (DNA), cell membranes, and cytoplasmic contents. However, the water content, determined from the wet-to-dry weight ratio, was 81.6% for natural tissue and 75.3% for dECM (Figure 3B), with no statistically significant difference, which indicated that the inherent composition and bio-architecture of the native extracellular matrix (ECM) would be preserved. Additionally, to quantify key extracellular matrix (ECM) components, the concentrations of collagen, elastin, and glycosaminoglycans (GAGs) were measured both before and after the decellularization process. Specifically, the collagen content (Figure 3C) decreased from 269.44 ± 3.07 μg/mg in native tissue to 162.25 ± 9.83 μg/mg in dECM. Similarly, the elastin content was reduced from 125.63 ± 1.13 μg/mg to 65.89±1.63 μg/mg, and the GAG content declined from 36.91 ± 1.63 μg/mg to 24.60 ± 1.46 μg/mg (Figure 3D,E).

It has been reported that to prevent potential immune responses, the DNA content in dECM-based biomaterials should be below 50 ng/mg dry weight [25]. The experimental data demonstrated that the decellularization protocol used in this study was highly efficacious, reducing the DNA content well below this critical threshold. Furthermore, quantitative analysis confirmed that, despite a significant reduction in their absolute quantities, essential ECM constituents were effectively preserved in the resulting dECM. Collectively, these results validate the efficacy of our decellularization protocol in achieving near-total clearance of cellular material while substantially maintaining the biochemical integrity of the native extracellular matrix.

### 3.2. Surface Morphology and Characterization of dECM/PCL Scaffolds

A series of scaffolds were fabricated via MEW to mimic the highly aligned structure of native extracellular matrix (ECM). The scaffolds were designed using Cura software with a fixed width of 400 μm and varying lengths of 400, 800 and 1200 μm, corresponding to aspect ratios of 1:1, 1:2, and 1:3, respectively (Figure 4A, top). To produce fine filaments, a 26-gauge nozzle (inner diameter: 250 μm) was used, which yielded fibers with diameters of 30 ± 5 μm that were directly printed onto a silicon wafer [26]. Thus, the printed scaffolds feature a highly porous (>90% porosity) architecture of aligned fibers.

This study was based on the premise that combining the bioactivity of dECM with the topographic guidance of aligned MEW scaffolds would enhance cell interactions [27]. Successful integration of a dECM coating was confirmed by SEM, which showed a uniform hydrogel layer on the PCL fibers, unlike the bare surface of the control (Figure 4A, down).

FTIR analysis provided further chemical evidence (Figure 4B). The spectrum of the pure PCL scaffold showed a characteristic carbonyl (C=O) stretching vibration at approximately 1724 cm^−1^. While NaOH pretreatment did not induce detectable chemical changes, coating with dECM resulted in a significant decrease in the intensity of the PCL-associated peak. Concurrently, new characteristic peaks emerged, corresponding to amide-I (1541 cm^−1^), amide-II (1644 cm^−1^), and N-H (3282 cm^−1^) vibrations, confirming the presence of dECM components on the composite scaffold. Importantly, these dECM signatures persisted after 24-h water immersion, confirming coating stability. Furthermore, the contact angle measurements confirmed the enhanced surface hydrophilicity resulting from the dECM coating (Figure 4C). The pristine PCL scaffold exhibited significant hydrophobicity, with a static contact angle of approximately 109.6°. After NaOH treatment and subsequent dECM coating, the scaffolds transformed into a hydrophilic surface and exhibited complete liquid absorption, demonstrating a transition to significant hydrophilicity. This enhanced surface wettability is expected to be more conducive to cell adhesion.

### 3.3. Mechanical Property of dECM/PCL Scaffolds

The mechanical properties of the scaffolds were primarily dependent on their architectural design. A pure PCL scaffold with a 1:1 aspect ratio was used as the control, exhibiting a tensile strength of 0.50 ± 0.05 MPa (Figure 5A) and an elastic modulus of 30 ± 6 kPa (Figure 5B). The incorporation of dECM onto this control scaffold resulted in a dECM/PCL composite that retained the original pore structure. Its tensile strength (0.55 ± 0.06 MPa) and elastic modulus (30 ± 4 kPa) were not significantly altered, indicating that the dECM coating had a minimal effect on the tensile properties. As the aspect ratio increased, a general trend of decreasing mechanical performance was observed. Although the 1:2 scaffold did not show a statistically significant reduction, the tensile modulus of the 1:3 scaffold decreased markedly, confirming that larger pores lead to inferior mechanical integrity. These mechanical characteristics are critically important, as substrate stiffness and strength directly influence cell adhesion, proliferation, and differentiation through mechano-transduction [28]. The robust mechanical properties of the 1:1 and 1:2 scaffolds likely provided a more stable and mechanically conducive microenvironment, thereby supporting the enhanced cell growth and alignment observed in our study.

### 3.4. Cell Viability and Morphology on dECM/PCL Scaffolds

The CCK-8 assay results (Figure 6A) indicated high cell viability across all groups after 1 h of incubation, with no significant cytotoxicity observed for either PCL or dECM/PCL scaffolds. Although initial cell attachment was limited on day 1, a substantial increase in adherent cells was evident by day 3. In the Figure 6B SEM images, the cells were observed at the grid top cross, line area and the grid side wall areas, which demonstrated that the dECM/PCL scaffolds were beneficial to cells proliferation and adhesion. The improved proliferation and adhesion performance can be attributed to the dECM coating, which introduced native extracellular matrix components that enhance scaffold bioactivity [29].

To better evaluate cell behavior, fluorescence microscopy was performed after 3 days of co-culture (Figure 7). While cells were able to adhere and proliferate on pure PCL scaffolds, enhanced adhesion and proliferation were observed on the dECM/PCL composite scaffolds, indicating that dECM decoration effectively improves the surface properties of the scaffolds to promote cellular activities. This finding was consistent with established literature, which demonstrated that surface modification with native ECM components significantly enhanced biocompatibility by providing innate cell-binding motifs. The synergistic combination of structural guidance from the aligned fibers, mechanical support from the composite, and biochemical signaling from the dECM created a favorable microenvironment that promoted superior cell alignment and growth, as evidenced by our experimental findings [30].

Furthermore, distinct morphological differences were noted. Cells on traditional culture dishes exhibited a non-polar, random distribution. In contrast, cells on the scaffolds preferentially adhered and elongated along the fibrous lines, with very few located in the pore spaces. This alignment was particularly evident on the 1:2 and 1:3 dECM/PCL scaffolds, where cells were predominantly found on the fibers. This observation aligned with the well-established concept of “contact guidance,” where topographical cues from the substrate directly dictated cell polarity and orientation by regulating cytoskeletal dynamics [31]. However, the 1:3 scaffold supported a lower cell density compared to the 1:1 and 1:2 variants, likely due to its larger pore size reducing the effective seeding density. Consequently, excessively large pores impair population growth by limiting cell–cell contacts and paracrine signaling [32].

Overall, the dECM-modified 1:1 and 1:2 scaffolds demonstrated significantly superior cell adhesion and proliferation compared to PCL controls. Critically, cells grew along the direction of the constituent fibers, demonstrating that the structural anisotropy of the scaffold can guide cell alignment. This successful guidance confirmed the principle that scaffold architecture was a powerful tool for controlling tissue-level organization, which was essential for engineering anisotropic tissues like maxillofacial skeletal muscle [33]. This suggested that the spatial distribution of cells can be directed by designing the scaffold’s architectural parameters, such as the dimensions and spacing of the lattice [34]. Our results collectively confirm that the integration of biochemical (dECM) and topographical (aligned fibers) cues is a potent strategy for developing advanced scaffolds that mimic the native tissue microenvironment.

## 4. Conclusions

By integrating the structural precision of MEW with the bioactivity of dECM, we successfully engineered scaffolds that mimic the anisotropic and biochemical characteristics of native soft tissues. The dECM-decorated PCL scaffolds promoted enhanced fibroblast adhesion, proliferation, and directional alignment, with pore architecture playing a critical role in modulating cellular behavior. This hybrid approach offers a promising strategy for developing biomimetic scaffolds tailored for oral and maxillofacial soft tissue repair and other regenerative applications. Further studies should investigate the effects of pore size, morphology, and other characteristics of MEW printed scaffolds on cell development and long-term cell survival rates.

## Figures and Tables

**Figure 1 polymers-17-03133-f001:**
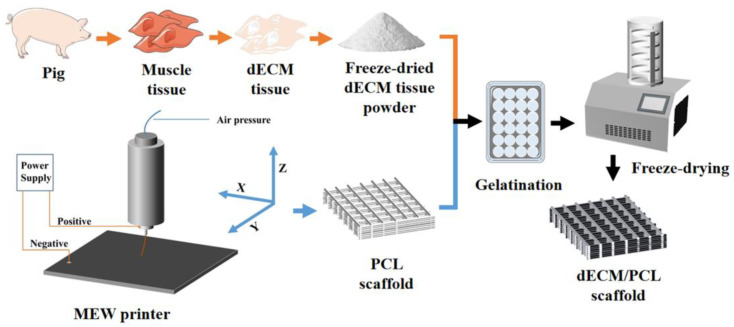
Schematic illustration of the fabrication process for the dECM/PCL Scaffold.

**Figure 2 polymers-17-03133-f002:**
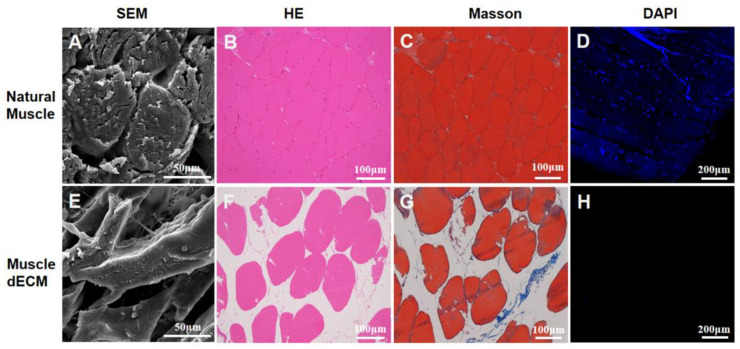
Decellularization of skeletal muscle tissue; (**A**,**E**) SEM images of natural and decellularized muscle tissue; (**B**–**D**) H&E, Masson trichrome, and DAPI staining images of natural muscle tissue; (**F**–**H**) H&E, Masson trichrome, and DAPI staining images of decellularized muscle tissue.

**Figure 3 polymers-17-03133-f003:**
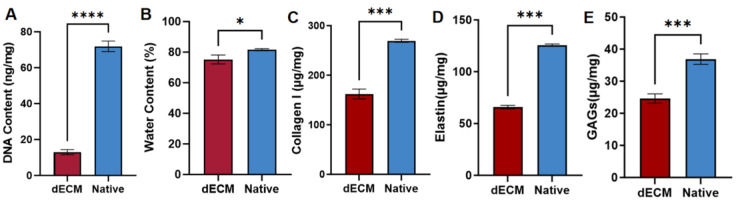
(**A**) DNA content; (**B**) Water content; (**C**) Collagen content; (**D**) Elastin content; and (**E**) GAG content of decellularized and native muscle tissue. (n = 3, * *p* ≤ 0.05, ** *p* ≤ 0.01, *** *p* ≤ 0.001, **** *p* ≤ 0.0001).

**Figure 4 polymers-17-03133-f004:**
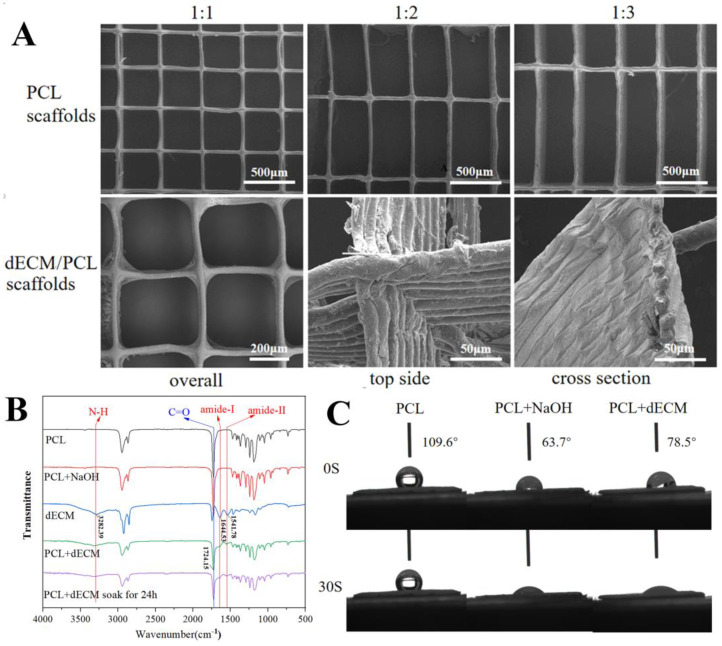
(**A**) SEM image; (**B**) FTIR analysis; and (**C**) Contact angle analysis of PCL scaffold and dECM/PCL scaffolds.

**Figure 5 polymers-17-03133-f005:**
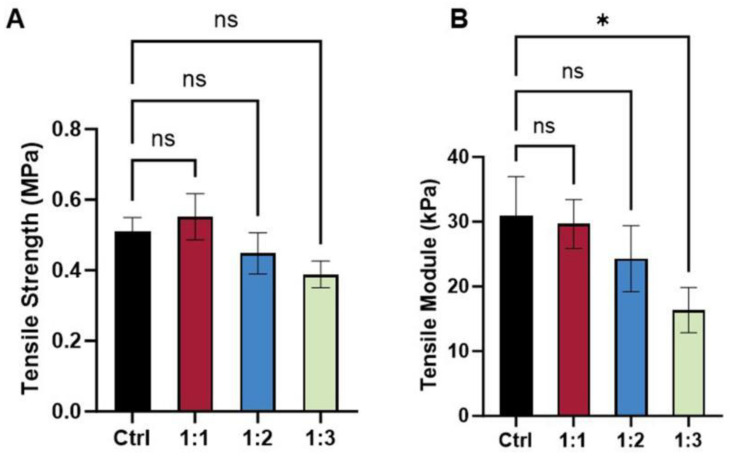
Mechanical properties of PCL and dECM/PCL scaffolds; (**A**) tensile strength; (**B**) tensile modulus. (n = 6, * *p* ≤ 0.05).

**Figure 6 polymers-17-03133-f006:**
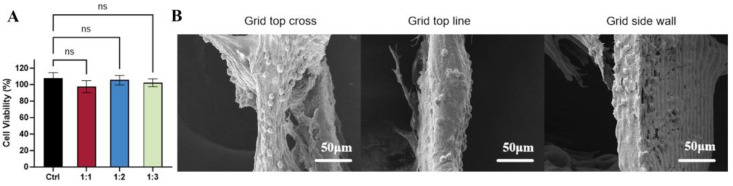
In vitro biological characteristics of PCL scaffolds and dECM/PCL scaffolds after 3 days of cultivation; (**A**) Cell survival rate in CCK8-confirmed scaffolds; (**B**) SEM images of cells adhere on dECM/PCL scaffolds. (n = 3).

**Figure 7 polymers-17-03133-f007:**
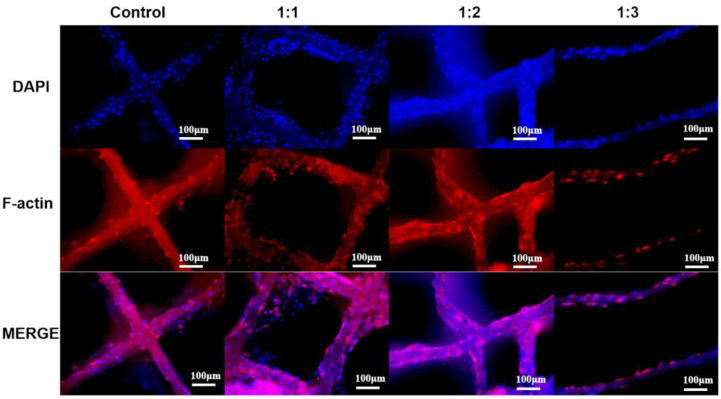
Fluorescent staining images of cells on different scaffolds after 3 days of culture (F-actin (red) and cell nuclei (blue).

## Data Availability

Data will be made available on request.

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
