# Peer review of "Modulating Cell–Scaffold Interaction via dECM-Decorated Melt Electrowriting PCL Scaffolds"

_polymers, 2025, doi:10.3390/polym17233133_

Round 1

Reviewer 1 Report

Comments and Suggestions for Authors

The authors have developed porous polycaprolactone (PCL) scaffolds with aligned microfibers and tunable pore architectures for cell-scaffold interaction investigation. This manuscript is suitable for polymers after revision.

  1. Is this system also suitable for other cells, except the  L929 fibroblasts?
  2. Please explain the effect of pore architectures on  L929 fibroblasts' behavior.
  3. Highlight the potential application area

Author Response

Comments 1: Is this system also suitable for other cells, except the L929 fibroblasts?

Response 1: Thank you for pointing this out. We agree with this comment. Yes, this system is certainly suitable for other cell types. In further studies, we will employ C2C12 cells, and previous research has utilized Mesenchymal Stem Cells (MSCs) and chondrocytes.

Comments 2: Please explain the effect of pore architectures on L929 fibroblasts' behavior.

Response 2: Thank you for pointing this out. We agree with this comment. Previous study-Brennan, C. M., K. F. Eichholz, and D. A. Hoey. "The effect of pore size within fibrous scaffolds fabricated using melt electrowriting on human bone marrow stem cell osteogenesis." Biomedical materials 14.6 (2019): 065016-has shown that the smaller pore sizes (e.g., 100μm) in melt electrowritten scaffolds enhance hMSC osteogenesis by promoting cell spreading, increasing local fiber stiffness, improving seeding efficiency, and boosting collagen and mineral deposition compared to larger pores (200μm and 300μm). In this literature, the scaffold pores were designed with a grid-like (square) morphology. Based on this structural characteristic, we hypothesized that rectangular pores could further promote anisotropic cell alignment. To test this hypothesis, we engineered a series of pore structures with aspect ratios ranging from 1:1 (square) to 1:3 (elongated rectangle). To improve cell-scaffold interaction, the surface of the scaffolds was functionalized with a decellularized extracellular matrix (dECM) coating, which was intended to provide a more bioactive interface and enhance initial cell attachment. Results demonstrated that scaffolds with aspect ratios of 1:1 and 1:2 significantly promoted cell adhesion and proliferation, suggesting a favorable microenvironment for cellular activities. In contrast, scaffolds with a 1:3 aspect ratio showed relatively poorer performance in terms of both cell attachment and proliferation.

Comments 3: Highlight the potential application area

Response 2: Thank you for pointing this out. We agree with this comment. For the potential application area, we expect the scaffold can be utilized as the scaffold of muscle tissue regeneration. We made revisions to the last paragraph of the introduction in the original manuscript, as follows: Therefore, this work aims to develop a biomimetic scaffold that synergistically integrates the topographic guidance of structurally anisotropic MEW PCL scaffolds with the biochemical signaling of dECM through a surface decoration approach. The ultimate goal is to systematically investigate cellular responses, including adhesion, proliferation, and alignment, to this dual-functional platform. To achieve this, dECM-decorated PCL scaffolds with controlled pore architectures (aspect ratios of 1:1, 1:2, and 1:3) were fabricated via melt electro writing (MEW) combined with a dip-gelation method. The biological performance of these composite scaffolds was thoroughly evaluated, focusing on their effects on fibroblast adhesion, viability, proliferation, and morphology. This study provides fundamental insights into the interplay between topographical and biochemical cues in guiding cell behavior, thereby laying a necessary foundation for the future development of advanced scaffolds for muscle tissue regeneration.

4. Response to Comments on the Quality of English Language

Point 1: (x)The English could be improved to more clearly express the research.

Response 1:  Thank you for pointing this out. We have modified several expressions in the revised manuscript.

Reviewer 2 Report

Comments and Suggestions for Authors

The paper has a good scientific soundness, and deserves to be published after major revisions.

The claimed novelty of this work lies in the combination of melt electrowriting (MEW)–printed, highly aligned PCL scaffolds with a bioactive coating derived from decellularized skeletal muscle extracellular matrix (dECM). In particular, the main innovation of this paper seems to be a very simple dip–gelation workflow to integrate skeletal-muscle dECM onto MEW-printed, highly aligned PCL, reported together with a controlled pore aspect-ratio sweep (1:1→1:3) and its effect on cell distribution and alignment in this specific configuration. While similar strategies combining MEW architectures and dECM-based coatings have been previously reported, the present study distinguishes itself by systematically exploring how pore geometry and biochemical coating jointly influence fibroblast alignment and proliferation. However, the novelty remains mainly methodological and incremental, rather than conceptual or mechanistic, as both MEW fabrication of anisotropic scaffolds and dECM functionalization of PCL are already well-established in the literature.

Here is the list of my observations about the manuscript.

Issue 1

The manuscript reports MEW-printed, aligned PCL scaffolds (square/rectangular pores with aspect ratios 1:1, 1:2, 1:3; ≈30 µm fibers; ~12×12×1 mm) subsequently “decorated” by dip–gelation with porcine skeletal-muscle dECM. The authors assess coating morphology, tensile properties, and short-term in-vitro response of L929 fibroblasts and conclude that combining anisotropic topology with dECM improves cell–scaffold interaction, with 1:1–1:2 pores performing best.

Issue 2

No quantitative coating metrics. Provide coating thickness/coverage (e.g., cross-sectional imaging + image analysis), mass gain per scaffold, contact angle/surface energy, and ideally XPS/FTIR to confirm biochemical deposition. Current SEM is qualitative only.

Issue 3

Stability of the coating is not assessed (e.g., after immersion, agitation, or cell culture). A retention test (protein content over time) and morphology after 3–7–14 days are necessary to show the coating persists under culture conditions.

Issue 4

Biochemical content of dECM post-processing. You quantify DNA (good), but key ECM constituents (collagen, GAGs) are not measured after pepsin/HCl digestion and neutralization; provide hydroxyproline (collagen) and DMMB (GAG) assays to confirm bioactive cargo survives the workflow.

Issue 5

The study lacks key experimental controls and relevant cell models. To distinguish biochemical from topographical effects, it should include PCL + NaOH, PCL + denatured dECM, and dECM hydrogel-only controls. Moreover, replacing or complementing L929 fibroblasts with human oral or tissue-specific cells would greatly improve the physiological relevance and strengthen the study’s claims.

Issue 6

Concerning mechanics, tensile tests were done on dry scaffolds; however, application is wet, cell-culture. Report wet mechanical properties, cyclic/dynamic modulus (DMA), and out-of-plane compliance relevant to soft tissue. Clarify sample size (n) for each condition and add representative stress–strain curves with shaded SD. Discuss whether dECM changes inter-fiber bonding or slip under load; if not, explain why mechanicals are unchanged despite the coating.

Issue 7

The study duration is short (≤3 days). Claims about regeneration should be toned down or supported with longer culture (e.g., 7–14 days), phenotypic markers (e.g., myogenic markers for skeletal muscle analogues), ECM deposition (fibronectin/collagen I immunostaining), and migration assays across pore geometries.

Issue 8

Considering figure numbering & consistency, it seems there are some inconsistencies (e.g., two “Figure 1” in different sections; check labels; ensure scale bars and magnifications on all micrographs).  

Author Response

Response to Reviewer 2 Comments

1. Summary

Thank you very much for taking the time to review this manuscript. Please find the detailed responses below and the corresponding revisions/corrections highlighted/in track changes in the re-submitted files. We highly appreciate these comments! Overall, the comments have been fair, encouraging and constructive. They are very helpful for revising and improving our paper. We have studied comments carefully and have made corrections which we hope meet with approval. Revised portions are marked in red in the paper. The main corrections in the paper and the response to the reviewer’s comments are as following:

2. Questions for General Evaluation

Reviewer’s Evaluation

Response and Revisions

Does the introduction provide sufficient background and include all relevant references?

Yes/Can be improved/Must be improved/Not applicable

we have improved the expression of the introduction.

Are all the cited references relevant to the research?

Yes/Can be improved/Must be improved/Not applicable

Is the research design appropriate?

Yes/Can be improved/Must be improved/Not applicable

We have added some experiment following the comments.

Are the methods adequately described?

Yes/Can be improved/Must be improved/Not applicable

We have modified following the comments.

Are the results clearly presented?

Yes/Can be improved/Must be improved/Not applicable

We have modified following the comments.

Are the conclusions supported by the results?

Yes/Can be improved/Must be improved/Not applicable

We have modified following the comments.

3. Point-by-point response to Comments and Suggestions for Authors

Comments 1: The manuscript reports MEW-printed, aligned PCL scaffolds (square/rectangular pores with aspect ratios 1:1, 1:2, 1:3; ≈30 µm fibers; ~12×12×1 mm) subsequently “decorated” by dip–gelation with porcine skeletal-muscle dECM. The authors assess coating morphology, tensile properties, and short-term in-vitro response of L929 fibroblasts and conclude that combining anisotropic topology with dECM improves cell–scaffold interaction, with 1:1–1:2 pores performing best.

Response 1: Thank you for pointing this out. We agree with this comment. This comment is a summary of the entire research content and does not require a reply.

Comments 2: No quantitative coating metrics. Provide coating thickness/coverage (e.g., cross-sectional imaging + image analysis), mass gain per scaffold, contact angle/surface energy, and ideally XPS/FTIR to confirm biochemical deposition. Current SEM is qualitative only.

Response 2: Agree. We have added the data on contact angle and FTIR. Please review the revised version as follows: A series of scaffolds were fabricated via MEW to mimic the highly aligned structure of native extracellular matrix (ECM). The scaffolds were designed using Cura software with a fixed width of 400μm and varying lengths of 400, 800 and 1200μm, corresponding to aspect ratios of 1:1, 1:2, and 1:3, respectively (Figure 4A, top). To produce fine filaments, a 26-gauge nozzle (inner diameter: 250 μm) was used, which yielded fibers with diameters of 30 ± 5 μm that were directly printed onto a silicon wafer [25]. Thus, the printed scaffolds feature a highly porous (>90% porosity) architecture of aligned fibers.

This study was based on the premise that combining the bioactivity of dECM with the topographic guidance of aligned MEW scaffolds would enhance cell interactions [26]. Successful integration of a dECM coating was confirmed by SEM, which showed a uniform hydrogel layer on the PCL fibers, unlike the bare surface of the control (Fig. 4A, down).

FTIR analysis provided further chemical evidence (Figure 4B). The spectrum of the pure PCL scaffold showed a characteristic carbonyl (C=O) stretching vibration at approximately 1724 cm-1. While NaOH pretreatment did not induce detectable chemical changes, coating with dECM resulted in a significant decrease in the intensity of the PCL-associated peak. Concurrently, new characteristic peaks emerged, corresponding to amide-I (1541 cm-1), amide-II (1644 cm-1), and N-H (3282 cm-1) vibrations, confirming the presence of dECM components on the composite scaffold. Importantly, these dECM signatures persisted after 24-hour water immersion, confirming coating stability. Furthermore, the contact angle measurements confirmed the enhanced surface hydrophilicity resulting from the dECM coating (Figure 4C). The pristine PCL scaffold exhibited significant hydrophobicity, with a static contact angle of approximately 109.6°. After NaOH treatment and subsequent dECM coating, the scaffolds transformed into a hydrophilic surface and exhibited complete liquid absorption, demonstrating a transition to significant hydrophilicity. This enhanced surface wettability is expected to be more conducive to cell adhesion.

Figure 4. (A) SEM image; (B) FTIR analysis; (C) Contact angle analysis of PCL scaffold and dECM/PCL scaffolds.

Comments 3: Stability of the coating is not assessed (e.g., after immersion, agitation, or cell culture). A retention test (protein content over time) and morphology after 3–7–14 days are necessary to show the coating persists under culture conditions.

Response 3: We appreciate your valuable and constructive suggestions. Due to the limited response time, we immersed the dECM-coated scaffolds in water at 37 °C for 24 hours to assess the stability of the coating. Subsequent FTIR analysis, as discussed in the Results section (Issue 2), confirmed that the characteristic peaks of dECM components remained detectable post-immersion, indicating the stability of the coating over this period.

Comments 4: Biochemical content of dECM post-processing. You quantify DNA (good), but key ECM constituents (collagen, GAGs) are not measured after pepsin/HCl digestion and neutralization; provide hydroxyproline (collagen) and DMMB (GAG) assays to confirm bioactive cargo survives the workflow.

Response 4: We appreciate your valuable and constructive suggestions. We have added the data on Collagen, elastin and GAGs quantitative analysis, actually these data have already been checked at the first time of the dECM preparation. Please review the revised version as follows:

To quantitatively evaluate decellularization efficiency, DNA content was analyzed relative to tissue dry weight. A significant decrease in DNA content was observed in the dECM (13.06 ± 1.47 ng/mg) compared to natural muscle (71.83 ± 3.10 ng/mg) (Figure 3A), confirming the effective removal of cellular material, such as nuclear components (DNA), cell membranes, and cytoplasmic contents. However, the water content, determined from the wet-to-dry weight ratio, was 81.6% for natural tissue and 75.3% for dECM (Figure 3B), with no statistically significant difference, which indicated that the inherent composition and bio-architecture of the native extracellular matrix (ECM) would be preserved. Additionally, to quantify key extracellular matrix (ECM) components, the concentrations of collagen, elastin, and glycosaminoglycans (GAGs) were measured both before and after the decellularization process. Specifically, the collagen content (Figure 3C) decreased from 269.44±3.07 μg/mg in native tissue to 162.25±9.83 μg/mg in dECM. Similarly, the elastin content was reduced from 125.63±1.13 μg/mg to 65.89±1.63 μg/mg, and the GAGs content declined from 36.91±1.63 μg/mg to 24.60±1.46 μg/mg (Figure 3D, E).

Figure 3. (A) DNA content; (B) Water content; (C) Collagen content; (D) Elastin content; and (E) GAG content of decellularized and native muscle tissue. (n=3, *p ≤ 0.05,** p ≤ 0.01, *** p ≤ 0.001, **** p ≤ 0.0001).

Comments 5: The study lacks key experimental controls and relevant cell models. To distinguish biochemical from topographical effects, it should include PCL + NaOH, PCL + denatured dECM, and dECM hydrogel-only controls. Moreover, replacing or complementing L929 fibroblasts with human oral or tissue-specific cells would greatly improve the physiological relevance and strengthen the study’s claims.

Response 5: We appreciate your valuable and constructive suggestions. While we fully agree with your perspective and plan to incorporate these recommendations into our future study designs, the current timeline unfortunately does not permit extensive modifications to the experimental framework. Thank you once again for your valuable suggestion.

Comments 6: Concerning mechanics, tensile tests were done on dry scaffolds; however, application is wet, cell-culture. Report wet mechanical properties, cyclic/dynamic modulus (DMA), and out-of-plane compliance relevant to soft tissue. Clarify sample size (n) for each condition and add representative stress–strain curves with shaded SD. Discuss whether dECM changes inter-fiber bonding or slip under load; if not, explain why mechanicals are unchanged despite the coating.

Response 6: We appreciate your valuable and constructive suggestions. In response, our findings indicate that the mechanical properties of the scaffold are primarily governed by its pore architecture rather than the presence of a surface coating. Specifically, while scaffolds with a 1:1 aspect ratio exhibited a marginally higher mechanical strength compared to the 1:2 and 1:3 designs, the differences were not statistically significant.

Regarding the potential influence of hydration, we would like to clarify our perspective. PCL is a polymer with a notably slow degradation rate, we hypothesize that short-term immersion in an aqueous environment (over a few days) is unlikely to induce substantial changes in its bulk mechanical properties. Therefore, we don't expect a big difference in the scaffold's mechanical performance between its dry and wet state. We postulate that the observed cellular responses are more likely influenced by surface chemical cues provided by the coating.

Nevertheless, we acknowledge the value of your point for ensuring comprehensive methodological rigor. We will certainly incorporate mechanical testing under wet conditions in subsequent phases of our research to fully substantiate this aspect.

Comments 7:  The study duration is short (≤3 days). Claims about regeneration should be toned down or supported with longer culture (e.g., 7–14 days), phenotypic markers (e.g., myogenic markers for skeletal muscle analogues), ECM deposition (fibronectin/collagen I immunostaining), and migration assays across pore geometries.

Response 7: We appreciate your valuable and constructive suggestions. This study employed L929 fibroblasts to investigate the combined effects of scaffold pore geometry and surface coatings on cellular behavior. Building on these findings, subsequent research will utilize C2C12 cells in longer-term cultures. This extended model will allow for the assessment of key parameters, including myogenic differentiation markers (e.g., MyoD, myogenin), ECM deposition (via immunostaining for fibronectin and collagen I), and cell migration across different pore geometries.

Comments 8: Considering figure numbering & consistency, it seems there are some inconsistencies (e.g., two “Figure 1” in different sections; check labels; ensure scale bars and magnifications on all micrographs).  

Response 8: We appreciate your valuable and constructive suggestions. In the revised version, we have made corrections to these errors. Thank you once again for your valuable suggestion.

4. Response to Comments on the Quality of English Language

Point 1: (x) The English is fine and does not require any improvement.

Response 1:

5. Additional clarifications

Reviewer 3 Report

Comments and Suggestions for Authors

The manuscript entitled “Modulating cell–scaffold interaction via dECM-decorated melt electro writing PCL scaffolds” describes the production of polycaprolactone (PCL) scaffolds using Melt Electro-Writing (MEW), decorated with decellularized extracellular matrix (dECM) from porcine skeletal muscle, and the evaluation of fibroblast behavior on these structures. The work is relevant to soft tissue engineering and falls within the scope of the journal Polymers. However, the manuscript requires substantial revisions before it can be considered for publication.

Recommendation: Minor revisions

Major comments

  1. Novelty and specific advancement

The introduction correctly frames the role of melt electro writing (MEW) in the fabrication of micro-structured scaffolds and mentions previous work on the use of decellularized matrices (dECM). However, the specific novelty of the present study is not explicitly stated. The authors indicate that they use dECM from porcine skeletal muscle to promote fibroblast adhesion and alignment, but they do not clarify whether the innovation lies in the source of the dECM, the geometry of the MEW lattices (ratios 1:1, 1:2, 1:3), or the combination of topography and bio-coating. We suggest adding a brief statement of novelty to the final paragraph of the introduction (p. 4) that clearly defines the advancement over previous studies. A small comparative table between MEW works with generic ECM coatings and the one proposed here would make the contribution of the work more evident.

  1. Decellularization and characterization of the matrix (Sections 2.1–2.2)

The decellularization protocol is clearly described and includes histological controls (H&E and DAPI) and quantification of residual DNA. However, no information is provided on the residual biochemical composition of the dECM (e.g., collagen, GAG, or total protein content). To ensure reproducibility and confirm the preservation of key structural components, it would be useful to add indicative data or, alternatively, cite previous work that has validated the same protocol with complete characterizations.

  1. Coating and surface characterization (Section 2.3)

The description of the coating is clear, but the experimental confirmation of the presence of dECM is only morphological. The SEM images show a variation in surface texture, but no direct evidence of composition. If possible, it is recommended to include a contact angle measurement (native PCL vs. dECM-coated), an FTIR spectrum with characteristic protein peaks (amide I/II ~1650/1550 cm⁻¹), or an approximate quantification of the amount of dECM deposited (µg cm⁻²). These data would clarify the success of the coating and improve the credibility of the biological conclusions.

  1. Mechanical properties 

Section 3.3 reports Young's moduli of ~30 kPa for MEW + dECM filaments. The authors state that “this stiffness falls within the range of soft tissues” but do not cite reference values. In the literature, skeletal muscle can have moduli in the order of hundreds of kPa or more (depending on the tissue). It would be useful to include in the discussion of the comparison between the modulus of the materials with that of the relevant biological material. Furthermore, the tests are performed under “dry” conditions; testing under hydrated conditions (e.g., in PBS at 37°C) would better simulate the culture environment and demonstrate mechanical stability over time (e.g., after 7 days in liquid).

Minor comments

Figures: The numbering of the figures is inconsistent: “Figure 1,” “Figure 2,” and “Figure 3” appear multiple times. Check and correct the numerical sequence, ensuring that captions and references in the text are consistent. Improve resolution and contrast, especially in fluorescent images.
Add (where absent) magnification information and scale bars in the captions of SEM and fluorescence micrographs to ensure consistency and readability between images.

Bibliography: Update the references section to include recent studies (2022–2024) on MEW–ECM scaffolds, anisotropic structures, and biofunctionalization strategies.

Author Response

Response to Reviewer 3 Comments

1. Summary

Thank you very much for taking the time to review this manuscript. Please find the detailed responses below and the corresponding revisions/corrections highlighted/in track changes in the re-submitted files. We highly appreciate these comments! Overall, the comments have been fair, encouraging and constructive. They are very helpful for revising and improving our paper. We have studied comments carefully and have made corrections which we hope meet with approval. Revised portions are marked in red in the paper. The main corrections in the paper and the response to the reviewer’s comments are as following:

2. Questions for General Evaluation

Reviewer’s Evaluation

Response and Revisions

Does the introduction provide sufficient background and include all relevant references?

Yes/Can be improved/Must be improved/Not applicable

we have improved the expression of the introduction.

Are all the cited references relevant to the research?

Yes/Can be improved/Must be improved/Not applicable

Is the research design appropriate?

Yes/Can be improved/Must be improved/Not applicable

We have added some experiment following the comments.

Are the methods adequately described?

Yes/Can be improved/Must be improved/Not applicable

We have modified following the comments.

Are the results clearly presented?

Yes/Can be improved/Must be improved/Not applicable

We have modified following the comments.

Are the conclusions supported by the results?

Yes/Can be improved/Must be improved/Not applicable

We have modified following the comments.

3. Point-by-point response to Comments and Suggestions for Authors

Comments 1: The introduction correctly frames the role of melt electro writing (MEW) in the fabrication of micro-structured scaffolds and mentions previous work on the use of decellularized matrices (dECM). However, the specific novelty of the present study is not explicitly stated. The authors indicate that they use dECM from porcine skeletal muscle to promote fibroblast adhesion and alignment, but they do not clarify whether the innovation lies in the source of the dECM, the geometry of the MEW lattices (ratios 1:1, 1:2, 1:3), or the combination of topography and bio-coating. We suggest adding a brief statement of novelty to the final paragraph of the introduction (p. 4) that clearly defines the advancement over previous studies. A small comparative table between MEW works with generic ECM coatings and the one proposed here would make the contribution of the work more evident.he manuscript reports MEW-printed, aligned PCL scaffolds (square/rectangular pores with aspect ratios 1:1, 1:2, 1:3; ≈30 µm fibers; ~12×12×1 mm) subsequently “decorated” by dip–gelation with porcine skeletal-muscle dECM. The authors assess coating morphology, tensile properties, and short-term in-vitro response of L929 fibroblasts and conclude that combining anisotropic topology with dECM improves cell–scaffold interaction, with 1:1–1:2 pores performing best.

Response 1: Thank you for pointing this out. We agree with this comment. We have added a brief statement in the last paragraph of the introduction, please review the revised version as follows:

Therefore, this work aims to develop a biomimetic scaffold that synergistically integrates the topographic guidance of structurally anisotropic MEW PCL scaffolds with the biochemical signaling of dECM through a surface decoration approach. The ultimate goal is to systematically investigate cellular responses, including adhesion, proliferation, and alignment, to this dual-functional platform. To achieve this, dECM-decorated PCL scaffolds with controlled pore architectures (aspect ratios of 1:1, 1:2, and 1:3) were fabricated via melt electro writing (MEW) combined with a dip-gelation method. The biological performance of these composite scaffolds was thoroughly evaluated, focusing on their effects on fibroblast adhesion, viability, proliferation, and morphology. This study provides fundamental insights into the interplay between topographical and biochemical cues in guiding cell behavior, thereby laying a necessary foundation for the future development of advanced scaffolds for muscle tissue regeneration.

Comments 2: Decellularization and characterization of the matrix (Sections 2.1–2.2)

The decellularization protocol is clearly described and includes histological controls (H&E and DAPI) and quantification of residual DNA. However, no information is provided on the residual biochemical composition of the dECM (e.g., collagen, GAG, or total protein content). To ensure reproducibility and confirm the preservation of key structural components, it would be useful to add indicative data or, alternatively, cite previous work that has validated the same protocol with complete characterizations.

Response 2: We appreciate your valuable and constructive suggestions. We have added the data on Collagen, elastin and GAGs quantitative analysis, actually these data have already been checked at the first time of the dECM preparation. Please review the revised version as follows:

To quantitatively evaluate decellularization efficiency, DNA content was analyzed relative to tissue dry weight. A significant decrease in DNA content was observed in the dECM (13.06 ± 1.47 ng/mg) compared to natural muscle (71.83 ± 3.10 ng/mg) (Figure 3A), confirming the effective removal of cellular material, such as nuclear components (DNA), cell membranes, and cytoplasmic contents. However, the water content, determined from the wet-to-dry weight ratio, was 81.6% for natural tissue and 75.3% for dECM (Figure 3B), with no statistically significant difference, which indicated that the inherent composition and bio-architecture of the native extracellular matrix (ECM) would be preserved. Additionally, to quantify key extracellular matrix (ECM) components, the concentrations of collagen, elastin, and glycosaminoglycans (GAGs) were measured both before and after the decellularization process. Specifically, the collagen content (Figure 3C) decreased from 269.44±3.07 μg/mg in native tissue to 162.25±9.83 μg/mg in dECM. Similarly, the elastin content was reduced from 125.63±1.13 μg/mg to 65.89±1.63 μg/mg, and the GAGs content declined from 36.91±1.63 μg/mg to 24.60±1.46 μg/mg (Figure 3D, E).

Figure 3. (A) DNA content; (B) Water content; (C) Collagen content; (D) Elastin content; and (E) GAG content of decellularized and native muscle tissue. (n=3, *p ≤ 0.05,** p ≤ 0.01, *** p ≤ 0.001, **** p ≤ 0.0001).

Comments 3: Coating and surface characterization (Section 2.3)

The description of the coating is clear, but the experimental confirmation of the presence of dECM is only morphological. The SEM images show a variation in surface texture, but no direct evidence of composition. If possible, it is recommended to include a contact angle measurement (native PCL vs. dECM-coated), an FTIR spectrum with characteristic protein peaks (amide I/II ~1650/1550 cm⁻¹), or an approximate quantification of the amount of dECM deposited (µg cm⁻²). These data would clarify the success of the coating and improve the credibility of the biological conclusions..

Response 3: Agree. We have added the data on contact angle and FTIR. Please review the revised version as follows: A series of scaffolds were fabricated via MEW to mimic the highly aligned structure of native extracellular matrix (ECM). The scaffolds were designed using Cura software with a fixed width of 400μm and varying lengths of 400, 800 and 1200μm, corresponding to aspect ratios of 1:1, 1:2, and 1:3, respectively (Figure 4A, top). To produce fine filaments, a 26-gauge nozzle (inner diameter: 250 μm) was used, which yielded fibers with diameters of 30 ± 5 μm that were directly printed onto a silicon wafer [25]. Thus, the printed scaffolds feature a highly porous (>90% porosity) architecture of aligned fibers.

This study was based on the premise that combining the bioactivity of dECM with the topographic guidance of aligned MEW scaffolds would enhance cell interactions [26]. Successful integration of a dECM coating was confirmed by SEM, which showed a uniform hydrogel layer on the PCL fibers, unlike the bare surface of the control (Fig. 4A, down).

FTIR analysis provided further chemical evidence (Figure 4B). The spectrum of the pure PCL scaffold showed a characteristic carbonyl (C=O) stretching vibration at approximately 1724 cm-1. While NaOH pretreatment did not induce detectable chemical changes, coating with dECM resulted in a significant decrease in the intensity of the PCL-associated peak. Concurrently, new characteristic peaks emerged, corresponding to amide-I (1541 cm-1), amide-II (1644 cm-1), and N-H (3282 cm-1) vibrations, confirming the presence of dECM components on the composite scaffold. Importantly, these dECM signatures persisted after 24-hour water immersion, confirming coating stability. Furthermore, the contact angle measurements confirmed the enhanced surface hydrophilicity resulting from the dECM coating (Figure 4C). The pristine PCL scaffold exhibited significant hydrophobicity, with a static contact angle of approximately 109.6°. After NaOH treatment and subsequent dECM coating, the scaffolds transformed into a hydrophilic surface and exhibited complete liquid absorption, demonstrating a transition to significant hydrophilicity. This enhanced surface wettability is expected to be more conducive to cell adhesion.

Figure 4. (A) SEM image; (B) FTIR analysis; (C) Contact angle analysis of PCL scaffold and dECM/PCL scaffolds.

Comments 4: Mechanical properties

Section 3.3 reports Young's moduli of ~30 kPa for MEW + dECM filaments. The authors state that “this stiffness falls within the range of soft tissues” but do not cite reference values. In the literature, skeletal muscle can have moduli in the order of hundreds of kPa or more (depending on the tissue). It would be useful to include in the discussion of the comparison between the modulus of the materials with that of the relevant biological material. Furthermore, the tests are performed under “dry” conditions; testing under hydrated conditions (e.g., in PBS at 37°C) would better simulate the culture environment and demonstrate mechanical stability over time (e.g., after 7 days in liquid).

Response 4: We appreciate your valuable and constructive suggestions. In response, our findings indicate that the mechanical properties of the scaffold are primarily governed by its pore architecture rather than the presence of a surface coating. Specifically, while scaffolds with a 1:1 aspect ratio exhibited a marginally higher mechanical strength compared to the 1:2 and 1:3 designs, the differences were not statistically significant.

Regarding the potential influence of hydration, we would like to clarify our perspective. PCL is a polymer with a notably slow degradation rate, we hypothesize that short-term immersion in an aqueous environment (over a few days) is unlikely to induce substantial changes in its bulk mechanical properties. Therefore, we don't expect a big difference in the scaffold's mechanical performance between its dry and wet state. We postulate that the observed cellular responses are more likely influenced by surface chemical cues provided by the coating.

Nevertheless, we acknowledge the value of your point for ensuring comprehensive methodological rigor. We will certainly incorporate mechanical testing under wet conditions in subsequent phases of our research to fully substantiate this aspect.

Comments 5: Figures: The numbering of the figures is inconsistent: “Figure 1,” “Figure 2,” and “Figure 3” appear multiple times. Check and correct the numerical sequence, ensuring that captions and references in the text are consistent. Improve resolution and contrast, especially in fluorescent images.

Add (where absent) magnification information and scale bars in the captions of SEM and fluorescence micrographs to ensure consistency and readability between images.

Response 5: We appreciate your valuable and constructive suggestions. In the revised version, we have made corrections to these errors. Thank you once again for your valuable suggestion. 

Comments 6: Bibliography: Update the references section to include recent studies (2022–2024) on MEW–ECM scaffolds, anisotropic structures, and biofunctionalization strategies.

Response 6: We appreciate your valuable and constructive suggestions. We have revised some expressions in the introduction and have focused on citing the research conducted from 2022 to 2024. Please review the revised version as follows:

Despite offering exceptional micro-architectural guidance, the inherent bio-inertness of synthetic MEW scaffolds often presents a ​less favorable​ environment for cell adhesion. To address this limitation, surface functionalization with bioactive coatings, such as gelatin and yeast-derived peptides, has emerged as an effective strategy to enhance the overall biocompatibility of MEW scaffolds by fostering more favorable cell-scaffold interactions [12,13].

Decellularized extracellular matrix (dECM) is a highly bioactive material derived from native tissues, capable of providing tissue-specific biochemical and physical cues to establish a supportive microenvironment that significantly promotes cell attachment, proliferation, and functional differentiation [14]. Harnessing this potential, Junka et al. combined a PCL scaffold with chondrocyte-derived dECM, calcium phosphate, and albumin to create a composite fibrocartilage-mimetic scaffold designed to emulate a woven bone precursor [15]. Sharma et al. fabricated 3D nanofiber scaffolds via gas-foaming expansion and functionalized them with cell-derived dECM to create a highly bioactive interface that enhances cellular alignment, reduces immune response, and improves tissue regeneration [16]. Despite this progress, the focus has predominantly been on scaffolds composed of nano- or sub-microscale fibers, such as those produced by electrospinning. The surface modification of melt electrowritten (MEW) scaffolds—which feature precisely controlled fiber diameters in the tens of micrometers—has received comparatively limited and unsystematic attention. An alternative integration strategy involves encapsulating the MEW scaffold within a hydrogel [17]. In this configuration, the MEW structure acts primarily as a mechanical support skeleton, while the hydrogel provides a bioactive environment for cell growth. In contrast to surface functionalization, this method often fails to leverage the finely tuned micro-architectural guidance of the MEW scaffold, as the critical topographical cues of the fibers are obscured by the surrounding hydrogel matrix.

12. Han, Y., Jia, B., Lian, M., et al. High-precision, gelatin-based, hybrid, bilayer scaffolds using melt electro-writing to repair cartilage injury[J]. Bioactive materials, 2021, 6(7), 2173-2186.

13. Mirzaei, M., Dodi, G., Gardikiotis, I., et al. 3D high-precision melt electro written polycaprolactone modified with yeast derived peptides for wound healing[J]. Biomaterials Advances, 2023, 149, 213361.

14. Cady, E., Orkwis, J. A., Weaver, R., et al. Micropatterning decellularized ECM as a bioactive surface to guide cell alignment, proliferation, and migration[J]. Bioengineering, 2020, 7(3), 102.

15. Junka, R., Zhou, X., Wang, W., et al. Albumin-coated polycaprolactone (PCL)–decellularized extracellular matrix (dECM) scaffold for bone regeneration[J]. ACS applied bio materials, 2022, 5(12), 5634-5644.

16. Sharma, N. S., Karan, A., Tran, H. Q., et al. Decellularized extracellular matrix-decorated 3D nanofiber scaffolds enhance cellular responses and tissue regeneration[J]. Acta Biomaterialia, 2024, 184, 81-97.

17. Galarraga, J. H., Locke, R. C., Witherel, C. E., et al.Fabrication of MSC-laden composites of hyaluronic acid hydrogels reinforced with MEW scaffolds for cartilage repair[J]. Biofabrication, 2021, 14(1), 014106.

4. Response to Comments on the Quality of English Language

Point 1: (x) The English could be improved to more clearly express the research.

Response 1: Thank you for pointing this out. We have modified several expressions in the revised manuscript.

5. Additional clarifications

Round 2

Reviewer 2 Report

Comments and Suggestions for Authors

Authors responded to my issues point by point, thus improving the quality of the manuscript.